# Equivariant MuZero

**Andreea Deac**[*]                                                                                              *deacandr@mila.quebec*
*Mila, Université de Montréal*

**Théophane Weber**                                                                                              *theophane@google.com*
*Google DeepMind*

**George Papamakarios**                                                                                          *gpapamak@google.com*
*Google DeepMind*

**Reviewed on OpenReview:** *https://openreview.net/forum?id=ExbGarTbLE*

## Abstract

Deep reinforcement learning has shown lots of success in closed, well-defined domains such as games (Chess, Go, StarCraft). The next frontier is real-world scenarios, where setups are numerous and varied. For this, agents need to learn the underlying environment dynamics, so as to robustly generalise to conditions that differ from those they were trained on. Model-based reinforcement learning algorithms, such as MuZero or Dreamer, aim to accomplish this by learning a world model. However, leveraging a world model has not yet consistently shown greater generalisation capabilities compared to model-free alternatives. In this work, we propose improving the data efficiency and generalisation capabilities of MuZero by explicitly incorporating the *symmetries* of the environment in its world-model architecture. We prove that, so long as the neural networks used by MuZero are equivariant to a particular symmetry group acting on the environment, the entirety of MuZero's action-selection algorithm will also be equivariant to that group. As such, Equivariant MuZero is guaranteed to behave symmetrically in symmetrically-transformed states, and will hence be more data-efficient when learning its world models. We evaluate Equivariant MuZero on procedurally-generated MiniPacman and on Chaser from the ProcGen suite: training on a set of mazes, and then testing on unseen rotated versions, demonstrating the benefits of equivariance. We verify that our improvements hold even when only some of the components of Equivariant MuZero obey strict equivariance, which highlights the robustness of our construction.

## 1 Introduction

Reinforcement learning (RL) is a potent paradigm for solving sequential decision making problems in a dynamically changing environment. Successful examples of its uses include game playing (Vinyals et al., 2019), drug design (Segler et al., 2018), robotics (Ibarz et al., 2021) and theoretical computer science (Fawzi et al., 2022). However, the generality of RL often leads to data inefficiency, poor generalisation and lack of safety guarantees. This is an issue especially in domains where data is scarce or difficult to obtain, such as medicine or human-in-the-loop scenarios.

Most RL approaches do not directly attempt to capture the regularities present in the environment. As an example, consider a grid-world: moving down in a maze is equivalent to moving left in the 90° clock-wise rotation of the same maze. Such equivalences can be formalised via Markov Decision Process homomorphisms (Ravindran, 2004; Ravindran & Barto, 2004), and while some works incorporate them (e.g. van der Pol et al., 2020; Rezaei-Shoshtari et al., 2022), most deep reinforcement learning agents would act differently in such equivalent states if they do not observe enough data. This becomes even more problematic when the number

---

[*]Work performed while the author was at Google DeepMind.

of equivalent states is large. One common example is 3D regularities, such as changing camera angles in robotic tasks.

In recent years, there has been significant progress in building deep neural networks that explicitly obey such regularities, often termed geometric deep learning (Bronstein et al., 2021). In this context, the regularities are formalised using symmetry groups and architectures are built by composing transformations that are equivariant to these symmetry groups (e.g. convolutional neural networks for the translation group, graph neural networks and transformers for the permutation group).

As we are looking to capture the symmetries present in an environment, a fitting place is within the framework of model-based RL (MBRL). MBRL leverages explicit world-models to forecast the effect of action sequences, either in the form of next-state or immediate reward predictions. These imagined trajectories are used to construct plans that optimise the forecasted returns. In the context of state-of-the-art MBRL agent MuZero (Schrittwieser et al., 2020), a Monte-Carlo tree search is executed over these world-models in order to perform action selection.

In this paper, we demonstrate that equivariance and MBRL can be effectively combined by proposing Equivariant MuZero (EqMuZero, shown in Figure 2), a variant of MuZero where equivariance constraints are enforced by design in its constituent neural networks. As MuZero does not use these networks directly to act, but rather executes a search algorithm on top of their predictions, it is not immediately obvious that the actions taken by the EqMuZero agent would obey the same constraints—is it guaranteed to produce a rotated action when given a rotated maze? One of our key contributions is a proof that guarantees this: as long as all neural networks are equivariant to a symmetry group, all actions taken will also be equivariant to that same symmetry group. Consequently, EqMuZero can be more data-efficient than standard MuZero, as it knows by construction how to act in states it has never seen before. On the practical side, we present a specific implementation of EqMuZero that is equivariant to 90° rotations by design, and we empirically verify its generalisation capabilities in two rotationally symmetric grid-worlds: procedurally-generated MiniPacman and the Chaser game in the ProcGen suite.

## 2 Background

### 2.1 Reinforcement Learning

The reinforcement learning problem is typically formalised as a Markov Decision Process $(S, A, P, R, \gamma)$ formed from a set of states $S$, a set of actions $A$, a discount factor $\gamma \in [0, 1]$, and two functions that model the outcome of taking action $a$ in state $s$: the transition distribution $P(s'|s, a)$ specifying the next state probabilities, and the reward function $R(s, a)$ specifying the reward. The aim is to learn a *policy*, $\pi(a|s)$, a function specifying (probabilities of) actions to take in state $s$, such that the agent maximises the (expected) cumulative reward $G(\texttt{tr}) = \sum_{t=0}^{t=T} \gamma^t R(s_t, a_t)$, where $\texttt{tr} = (s_0, a_0, s_1, a_1, \ldots, s_T, a_T)$ is the trajectory taken by the agent starting in the initial state $s_0$ and following the policy to decide $a_t$ based on $s_t$.

### 2.2 MuZero

Reinforcement learning agents broadly fall into two categories: *model-free* and *model-based*. The specific agent we extend here, MuZero (Schrittwieser et al., 2020), is a model-based agent for deterministic environments (where $P(s'|s, a) = 1$ for exactly one $s'$ for all $s \in S$ and $a \in A$). MuZero relies on several neural-network components that are composed to create a *world model*, which estimates the unseen processes in the MDP.

The neural networks are:

- The *state encoder*, $h : S \to Z$, which embeds states into a latent space $Z$ (e.g. $Z = \mathbb{R}^k$).

- The *action encoder*, $h_A : A \to Z_A$, which embeds actions into a latent space $Z_A$ (e.g. $Z_A = \mathbb{R}^k$).

- The *transition function*[1], $\tau : Z \times A \to Z$, which predicts embeddings of next states.

---

[1]Note that both the transition and reward functions will implicitly call the action encoder, $h_A$, in order to appropriately embed the input action. We expand on this in Section 3.1.

- The *reward function*, $\rho : Z \times A \to \mathbb{R}$, which predicts the immediate expected reward after taking an action in a particular state.

- The *value function*, $v : Z \to \mathbb{R}$, which predicts the value (expected cumulative reward) from a given state.

- The *policy function*, $p : Z \to [0, 1]^{|A|}$, which predicts the probability of taking each action from the current state. These probabilities should add up to 1: $\sum_{a \in A} p(a|\mathbf{z}) = 1$.

Using these, MuZero computes its corresponding transition, reward, value and policy models respectively through composition of neural networks.

To plan its next action, MuZero executes a Monte Carlo tree search (MCTS) over many simulated trajectories, generated using the above models. We use superscript notation to denote information belonging to planning such as $z^0$ being the hidden representation of the root state in a MCTS step, and subscripts for interacting with the environment such as $a_t$ being the action taken at time $t$. Moreover, please note that we use $s$ for input observation and $z$ for latent space representations, while the original MuZero paper (Schrittwieser et al., 2020) uses $o$ and $s$ respectively.

MuZero has demonstrated state-of-the-art capabilities over a variety of deterministic or near-deterministic environments, such as Go, Chess, Shogi and Atari, and has been successfully applied to real-world domains such as video compression (Mandhane et al., 2022). Although here we focus on MuZero for deterministic environments, we note that extensions to stochastic environments also exist (Antonoglou et al., 2021) and are an interesting target for future work.

### 2.3 Groups, Representations and Symmetries

A *group* $(\mathfrak{G}, \circ)$ is a set $\mathfrak{G}$ equipped with a *composition* operation $\circ : \mathfrak{G} \times \mathfrak{G} \to \mathfrak{G}$ (written concisely as $\mathfrak{g} \circ \mathfrak{h} = \mathfrak{g}\mathfrak{h}$), satisfying the following axioms:

- *Associativity*: $(\mathfrak{g}\mathfrak{h})\mathfrak{l} = \mathfrak{g}(\mathfrak{h}\mathfrak{l})$ for all $\mathfrak{g}, \mathfrak{h}, \mathfrak{l} \in \mathfrak{G}$.

- *Identity*: there exists a unique $\mathfrak{e} \in \mathfrak{G}$ satisfying $\mathfrak{e}\mathfrak{g} = \mathfrak{g}\mathfrak{e} = \mathfrak{g}$ for all $\mathfrak{g} \in \mathfrak{G}$.

- *Inverse*: for every $\mathfrak{g} \in \mathfrak{G}$ there exists a unique $\mathfrak{g}^{-1} \in \mathfrak{G}$ such that $\mathfrak{g}\mathfrak{g}^{-1} = \mathfrak{g}^{-1}\mathfrak{g} = \mathfrak{e}$.

Groups are a natural way to describe *symmetries*: object transformations that leave the object unchanged.

Since group elements are just abstract set elements, in order to reason about them as transformations, we define the notion of a *group action* $\texttt{act} : \mathfrak{G} \times \Omega \to \Omega$, where $\Omega$ is the space of the input data. For example, if $\Omega = \mathbb{Z}_n^2$ refers to the pixels of an image and $\mathfrak{G} = \mathbb{Z}_n^2$ refers to circular translations, the group action $\texttt{act}$ translates each pixel accordingly:

$$\texttt{act}((a, b), (u, v)) = ((u + a) \bmod n, (v + b) \bmod n) \tag{1}$$

where $(a, b) \in \mathfrak{G}$ specifies the translation operation and $(u, v) \in \Omega$ is the pixel being translated.

In most cases of interest, the group actions will be linear transformations of the data in $\Omega$. We can thus reason about them in the context of linear algebra by using their *real representations*: functions $\rho_{\mathcal{V}} : \mathfrak{G} \to \mathbb{R}^{N \times N}$ that give, for every group element $\mathfrak{g} \in \mathfrak{G}$, a real matrix demonstrating how this element *acts* on a vector space $\mathcal{V}$. For example, for the rotation group $\mathfrak{G} = \mathrm{SO}(n)$, the representation $\rho_{\mathcal{V}}$ would provide an appropriate $n \times n$ rotation matrix for each rotation $\mathfrak{g}$. We perform the group action by multiplying the matrices corresponding to the representation and our data.

Note that we use $\rho$ to denote both a group representation and the reward function of MuZero. However, they should be easy to distinguish as the representation function will always be applied to group elements such as $\mathfrak{g}$, whereas the reward function will be applied to neural network embeddings $\mathbf{z}$.

## 2.4 Equivariance and Invariance

As symmetries are assumed to not change the essence of the data they act on, we would like to construct neural networks that adequately represent such symmetry-transformed inputs. Assume we have a neural network $f : \mathcal{X} \to \mathcal{Y}$, mapping between vector spaces $\mathcal{X}$ and $\mathcal{Y}$, and that we would like this network to respect the symmetries within a group $\mathfrak{G}$. Then we can impose the following condition, for all group elements $\mathfrak{g} \in \mathfrak{G}$ and inputs $\mathbf{x} \in \mathcal{X}$:

$$f(\rho_{\mathcal{X}}(\mathfrak{g})\mathbf{x}) = \rho_{\mathcal{Y}}(\mathfrak{g})f(\mathbf{x}). \tag{2}$$

This condition is known as $\mathfrak{G}$-*equivariance*: for any group element, it does not matter whether we act with it on the input or on the output of the function $f$—the end result is the same. A special case of this, $\mathfrak{G}$-*invariance*, is when the output representation is trivial ($\rho_{\mathcal{Y}}(\mathfrak{g}) = \mathbf{I}$):

$$f(\rho_{\mathcal{X}}(\mathfrak{g})\mathbf{x}) = f(\mathbf{x}). \tag{3}$$

In geometric deep learning, equivariance to reflections, rotations, translations and permutations has been of particular interest (Bronstein et al., 2021). For the specific context of reinforcement learning and MuZero, the possible inputs to our neural networks may be both states and actions. Therefore, when further discussing these conditions, note that $\mathbf{x}$ may refer to either state or action representations, or both at once.

Generally speaking, there are three ways to obtain an equivariant model: a) data augmentation, b) data canonicalisation and c) specialised architectures. Data augmentation creates additional training data by applying group elements $\mathfrak{g}$ to input/output pairs $(\mathbf{x}, \mathbf{y})$—equivariance is encouraged by training on the transformed data and/or minimising auxiliary losses such as $\|\rho_{\mathcal{Y}}(\mathfrak{g})f(\mathbf{x}) - f(\rho_{\mathcal{X}}(\mathfrak{g})\mathbf{x})\|$. Data augmentation can be simple to apply, but it results in only approximately equivariant models. Data canonicalisation requires a method to standardise the input, such as breaking the translation symmetry for molecular representation by centering the atoms around the origin (Musil et al., 2021)—however, in many cases such a canonical transformation may not exist. Specialised architectures have the downside of being harder to build, but they can guarantee exact equivariance—as such, they reduce the search space of functions, potentially reducing the number of parameters and increasing training efficiency.

Bronstein et al. (2021) discuss many ways in which one can construct state-of-the-art specialised architectures, from rudimentary approaches relying on exhaustive view generation (Cohen & Welling, 2016a), to more efficient approaches like Steerable CNNs (Cohen & Welling, 2016b; Weiler & Cesa, 2019), all the way to advanced approaches preserving gauge equivariance over manifolds and meshes (De Haan et al., 2020). All of these approaches can be leveraged to build constituent networks in our Equivariant MuZero framework.

## 2.5 Equivariance in RL and Related Work

There has been previous work at the intersection of reinforcement learning and equivariance. While leveraging multi-agent symmetries was repeatedly shown to hold promise (van der Pol et al., 2021; Muglich et al., 2022), of particular interest to us are the symmetries emerging from the environment, in a single-agent scenario.

Related work in this space can be summarised by the commutative diagram in Figure 1. This diagram represents various transformations we might want to make on data that originates from RL, as arrows. Some of these arrows are neural network layers (such as $\text{Enc} : S \to Z$), some are action executions (such as $a : S \to S$), while others are symmetry transformations (such as $\rho_{\mathcal{Z}}(\mathfrak{g}) : Z \to Z$). In a commutative diagram, each pair of paths connecting the same start and endpoint specifies a *mathematical constraint* that the compositions of arrows in these two paths must be the same transformation.

For example, in Figure 1, consider the front-facing square of the lower cube, i.e., paths $S \xrightarrow{\rho_S(\mathfrak{g})} S \xrightarrow{\text{Enc}} Z$ and $S \xrightarrow{\text{Enc}} Z \xrightarrow{\rho_{\mathcal{Z}}(\mathfrak{g})} Z$. As these two paths both start in the same point and end in the same point, we recover the previously discussed $\mathfrak{G}$-equivariance condition on Enc:

$$\text{Enc}(\rho_S(\mathfrak{g})s) = \rho_{\mathcal{Z}}(\mathfrak{g})\text{Enc}(s). \tag{4}$$

When considering only the cube at the bottom, we recover Park et al. (2022)—a supervised learning task where a latent transition model $T$ learns to predict the next state embedding. They show that if $T$ is

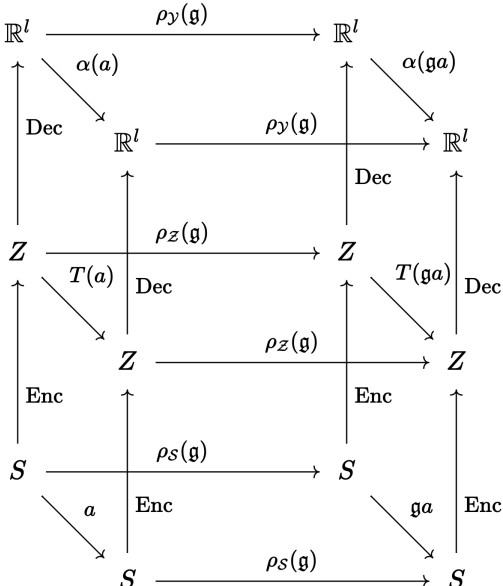

Figure 1: Commutative diagram of symmetries in RL. State transitions due to an action $a$ are back-to-front, transformations due to a symmetry $\mathfrak{g}$ are left-to-right, state encoding and decoding by the model is bottom-to-top.

equivariant, the encoder can pick up the symmetries of the environment even if it is not fully equivariant by design. Mondal et al. (2022) build a model-free agent by combining an equivariant-by-design encoder and enforcing the remaining equivariances via regularisation losses. They also consider the invariance of the reward, captured in Figure 1 by taking the decoder to be the reward model and $l = 1$. The work of van der Pol et al. (2020) can be described by having the value model as the decoder, while the work of Wang et al. (2022) has the policy model as the decoder and $l = |A|$.

While our work is the first to study the interplay of equivariance and Monte-Carlo tree search (MCTS)-backed MBRL algorithms such as MuZero, we would like to point out three recent related works that are generally inspired by exploiting symmetries in model-based planning. Firstly, SymPlan (Zhao et al., 2022) takes inspiration from the symmetries present in the Value Iteration algorithm (Bellman, 1966), to extend Value Iteration Networks (Tamar et al., 2016) with additional equivariance constraints. Note that, while there exists a MBRL inspiration in this work, it is still an *implicit planner*, i.e. an end-to-end agent model without any explicit invocation of model-based methods. Secondly, EDGI (Brehmer et al., 2023) is a recent extension of diffusion-based planning (Janner et al., 2022) that incorporates group equivariance constraints. As planning with diffusion naturally occurs in a high-dimensional space with rich geometry, imposing additional spatial constraints on it is a natural direction to follow. Lastly, in a work concurrent to ours, Zhao et al. (2023) study how various path-planning algorithms—e.g., model predictive control—behave under symmetries, and derive arguments that are similar in spirit to our theoretical analysis of MuZero under equivariance.

## 3 Equivariant MuZero

This section describes Equivariant MuZero (EqMuZero), a variant of MuZero whose constituent neural networks (encoder, transition model, reward model, value model and policy model) are equivariant by design. Section 3.1 describes a specific implementation of EqMuZero, where the equivariance is with respect to the 4-element cyclic group $C_4$, that is, the group of 90° rotations. Section 3.2 formally proves that if the components of EqMuZero are equivariant with respect to any symmetry group $\mathfrak{G}$, the action selection will also be equivariant with respect to $\mathfrak{G}$.

### 3.1   Implementation of $C_4$-equivariant MuZero

We present a concrete instance of EqMuZero for the 4-element cyclic group $C_4$, and we describe how its various components can be designed to obey $C_4$-equivariance (Figure 2). The $C_4$ group is applicable, for example, in rotationally symmetric grid-worlds, where the elements of the group represent rotating the grid-world by all four possible multiples of $90°$, that is, $C_4 = \{\mathfrak{e}, \mathfrak{r}_{90°}, \mathfrak{r}_{180°}, \mathfrak{r}_{270°}\}$. Such grid-worlds are symmetric with respect to $90°$ rotations, in the sense that moving down in a grid-world map is the same as moving left in the $90°$ clock-wise rotated version of the same map. As our exemplar, we will use the G-CNN architecture from Cohen & Welling (2016a), which is among the earliest prominent architectures that directly supports equivariance to small finite groups, such as $C_4$.

In what follows we assume the environment is a grid-world for concreteness, however we note that our EqMuZero implementation is directly applicable to any environment with the same symmetry structure. Let $\mathbf{s}$ denote the state of the grid-world represented as a 2D array (i.e. an image). For simplicity, we assume there are only four directional movement actions in the environment, denoted by $A = \{\rightarrow, \downarrow, \leftarrow, \uparrow\}$. Any additional non-movement actions (such as the "do nothing" action) can be included without difficulty. We assume that the $C_4$ group acts on states and actions in the obvious way: $\rho_{\mathcal{S}}(\mathfrak{r}_{90°})\mathbf{s} = \mathbf{R}_{90°}\mathbf{s}$ is the grid-world state rotated clock-wise by $90°$, and $\rho_{\mathcal{A}}(\mathfrak{r}_{90°}) \rightarrow = \downarrow$.

**State and action encoders**   To enforce $C_4$-equivariance in the state encoder, we first need to specify the effect of rotations on the latent state $\mathbf{z}$. In our implementation, the latent state consists of 4 equally shaped arrays, $\mathbf{z} = (\mathbf{z}_1, \mathbf{z}_2, \mathbf{z}_3, \mathbf{z}_4)$, and we prescribe that a $90°$ clock-wise rotation manifests as a cyclical permutation: $\rho_{\mathcal{Z}}(\mathfrak{r}_{90°})\mathbf{z} = (\mathbf{z}_2, \mathbf{z}_3, \mathbf{z}_4, \mathbf{z}_1)$. Then, our equivariant encoder embeds state $\mathbf{s}$ as follows:

$$h_{\mathsf{eq}}(\mathbf{s}) = (h(\mathbf{s}), h(\mathbf{R}_{90°}\mathbf{s}), h(\mathbf{R}_{180°}\mathbf{s}), h(\mathbf{R}_{270°}\mathbf{s})) = (\mathbf{z}_1, \mathbf{z}_2, \mathbf{z}_3, \mathbf{z}_4) \tag{5}$$

where $h$ is an arbitrary neural network (we use a CNN). Similarly, our equivariant action-encoder embeds action $a$ as follows:

$$h_{A\text{-}\mathsf{eq}}(a) = (h_A(a), h_A(\rho_{\mathcal{A}}(\mathfrak{r}_{90°})a), h_A(\rho_{\mathcal{A}}(\mathfrak{r}_{180°})a), h_A(\rho_{\mathcal{A}}(\mathfrak{r}_{270°})a)) = (\mathbf{z}_{a;1}, \mathbf{z}_{a;2}, \mathbf{z}_{a;3}, \mathbf{z}_{a;4}) \tag{6}$$

where $h_A$ is an MLP.

**Transition model**   We propose two ways of building a $C_4$-equivariant transition model. The first one works by maintaining the structure in the latent space:

$$\tau_{\mathsf{eq}}(\mathbf{z}, a) = (\tau(\mathbf{z}_1, h_{A\text{-}\mathsf{eq}}(a)_1), \tau(\mathbf{z}_2, h_{A\text{-}\mathsf{eq}}(a)_2), \tau(\mathbf{z}_3, h_{A\text{-}\mathsf{eq}}(a)_3), \tau(\mathbf{z}_4, h_{A\text{-}\mathsf{eq}}(a)_4)), \tag{7}$$

where $h_{A\text{-}\mathsf{eq}}(a)_1$ represents the first element of the tuple obtained from encoding action $a$. The network $\tau$ is arbitrary; in our implementation, we broadcast the output of $h_A$ across all pixels of $h$'s output, followed by a ResNet. This equation satisfies $C_4$-equivariance, that is, $\tau_{\mathsf{eq}}(\mathbf{R}_{90°}\mathbf{s}, \rho_{\mathcal{A}}(\mathfrak{r}_{90°})a) = \rho_{\mathcal{Z}}(\mathfrak{r}_{90°})\tau_{\mathsf{eq}}(\mathbf{s}, a)$.

Our second transition model is less constrained but more involved, as it allows components of $\mathbf{z}$ to *interact*, while still retaining $C_4$-equivariance:

$$\tau_{\mathsf{eq}}(\mathbf{z}, a) = (\tau(\tilde{\mathbf{z}}_1, \tilde{\mathbf{z}}_2, \tilde{\mathbf{z}}_3, \tilde{\mathbf{z}}_4), \tau(\tilde{\mathbf{z}}_2, \tilde{\mathbf{z}}_3, \tilde{\mathbf{z}}_4, \tilde{\mathbf{z}}_1), \tau(\tilde{\mathbf{z}}_3, \tilde{\mathbf{z}}_4, \tilde{\mathbf{z}}_1, \tilde{\mathbf{z}}_2), \tau(\tilde{\mathbf{z}}_4, \tilde{\mathbf{z}}_1, \tilde{\mathbf{z}}_2, \tilde{\mathbf{z}}_3)). \tag{8}$$

For brevity, $\tilde{\mathbf{z}}_1$ denotes the pair $(\mathbf{z}_1, h_{A\text{-}\mathsf{eq}}(a)_1)$. Note that, in this version, $\tau$ takes more arguments than before. In our experiments, we use the more constrained variant for MiniPacman, and the less constrained variant for Chaser, as more data is available for the latter.

**Policy model**   We make a $C_4$-equivariant policy by combining state and action embeddings from all four latents as follows:

$$p_{\mathsf{eq}}(a \mid \mathbf{z}) = \frac{p(a \mid \mathbf{z}_1) + p(\rho_{\mathcal{A}}(\mathfrak{r}_{90°})a \mid \mathbf{z}_2) + p(\rho_{\mathcal{A}}(\mathfrak{r}_{180°})a \mid \mathbf{z}_3) + p(\rho_{\mathcal{A}}(\mathfrak{r}_{270°})a \mid \mathbf{z}_4)}{4}. \tag{9}$$

It is straightforward to verify that $\sum_{a \in A} p_{\mathsf{eq}}(a \mid \mathbf{z}) = 1$, i.e. $p_{\mathsf{eq}}(\cdot \mid \mathbf{z})$ is properly normalised, and that $p_{\mathsf{eq}}(\rho_{\mathcal{A}}(\mathfrak{r}_{90°})a \mid \rho_{\mathcal{Z}}(\mathfrak{r}_{90°})\mathbf{z}) = p_{\mathsf{eq}}(a \mid \mathbf{z})$, i.e. it satisfies $C_4$-equivariance.

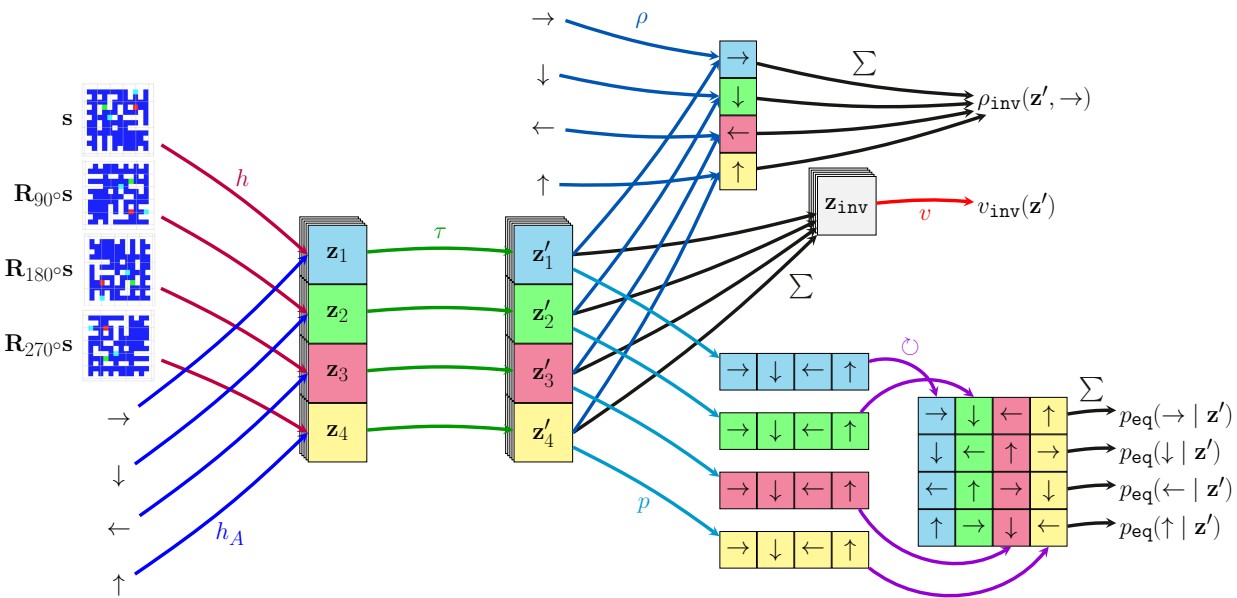

Figure 2: Architecture of Equivariant MuZero, where $h$, $h_A$ are encoders, $\tau$ is the transition model, $\rho$ is the reward model, $v$ is the value model and $p$ is the policy predictor. Each colour represents an element of the $C_4$ group $\{\mathfrak{e}, \mathfrak{r}_{90°}, \mathfrak{r}_{180°}, \mathfrak{r}_{270°}\}$ applied to the input (environment state and action).

**Reward and value models** Lastly, the reward and value networks ($\rho_{\texttt{inv}}$, $v_{\texttt{inv}}$) should be $C_4$-invariant. We can satisfy this constraint by *aggregating* the latent space with any $C_4$-invariant function, such as sum, average or max. Here we use summation:

$$\rho_{\texttt{inv}}(\mathbf{z}, a) = \rho(\mathbf{z}_1, a) + \rho(\mathbf{z}_2, \rho_{\mathcal{A}}(\mathfrak{r}_{90°})a) + \rho(\mathbf{z}_3, \rho_{\mathcal{A}}(\mathfrak{r}_{180°})a) + \rho(\mathbf{z}_4, \rho_{\mathcal{A}}(\mathfrak{r}_{270°})a) \tag{10}$$

$$v_{\texttt{inv}}(\mathbf{z}) = v(\mathbf{z}_1) + v(\mathbf{z}_2) + v(\mathbf{z}_3) + v(\mathbf{z}_4). \tag{11}$$

Composing the equivariant components described above (Equations 5–10), we construct the end-to-end equivariant EqMuZero agent, displayed in Figure 2.

While the specific implementation presented here represents a valid EqMuZero architecture for the $C_4$ symmetry group, we reiterate that it is only an exemplar. If we were interested in expanding to a broader symmetry group such as $D_4$—which includes eight elements, spanning both rotations and reflections—we could extend our representations to include all eight transformed views, or utilise a more memory-efficient approach of Steerable CNNs (Cohen & Welling, 2016b; Weiler & Cesa, 2019; Zhao et al., 2022) or explicit symmetrisation (van der Pol et al., 2020). In fact, we now prove a theorem stating that, no matter the symmetry group of choice or the method used to ensure equivariance to it (Bronstein et al., 2021), so long as equivariance is guaranteed, EqMuZero will select actions in a way that respects the underlying symmetries.

### 3.2 Proof of Action Equivariance for Arbitrary Symmetry Groups

As discussed in Section 2.2, MuZero does not use its constituent networks directly for acting, but it uses them to perform a Monte-Carlo tree search (MCTS), from which an action is selected. Therefore, even if MuZero's constituent networks are fully equivariant, it is not obvious that the action selection is too. In other words, are we guaranteed that EqMuZero will behave in an equivariant manner, that is, will it always perform a rotated action in a rotated grid-world? The following theorem shows that the answer is yes.

**Theorem 1** *If all the relevant neural networks used by MuZero are $\mathfrak{G}$-equivariant, the proposed EqMuZero agent will select actions in a $\mathfrak{G}$-equivariant manner, that is for every state $s \in S$ and for every $\mathfrak{g} \in \mathfrak{G}$, if EqMuZero selects action $a$ while in $s$, then it must select $\rho_{\mathcal{A}}(\mathfrak{g})a$ while in $\rho_{\mathcal{S}}(\mathfrak{g})s$.*

**Proof.** Assume that the EqMuZero agent uses the following equivariant neural networks: $h_{\text{eq}}$ for the encoder (which also includes the action encoder for simplicity), $\tau_{\text{eq}}$ for the transition model, $p_{\text{eq}}$ for the policy model, $v_{\text{inv}}$ for the value model and $\rho_{\text{inv}}$ for the reward function. By assumption, $h_{\text{eq}}, \tau_{\text{eq}}$ and $p_{\text{eq}}$ are $\mathfrak{G}$-equivariant, and $v_{\text{inv}}$ and $\rho_{\text{inv}}$ are $\mathfrak{G}$-invariant.

MuZero relies on multi-step rollouts using its internal models. Specifically, it requires simulating the reward, value, policy and transition models $n$ steps in the future, starting from state $s$ and simulating the action sequence $a^1, a^2, \ldots, a^n$. We can simulate these rollouts as the following compositions of neural networks:

$$
\begin{aligned}
r^n &= \rho_{\text{inv}}(\tau_{\text{eq}}(\cdots \tau_{\text{eq}}(\tau_{\text{eq}}(h_{\text{eq}}(s), a^1), a^2) \cdots, a^{n-1}), a^n) \\
v^n &= v_{\text{inv}}(\tau_{\text{eq}}(\cdots \tau_{\text{eq}}(\tau_{\text{eq}}(h_{\text{eq}}(s), a^1), a^2) \cdots, a^n)) \\
\mathbf{p}^n &= p_{\text{eq}}(\tau_{\text{eq}}(\cdots \tau_{\text{eq}}(\tau_{\text{eq}}(h_{\text{eq}}(s), a^1), a^2) \cdots, a^n)) \\
z^n &= \tau_{\text{eq}}(\cdots \tau_{\text{eq}}(\tau_{\text{eq}}(h_{\text{eq}}(s), a^1), a^2) \cdots, a^n).
\end{aligned}
\tag{12}
$$

As these rollout simulations are computed by composing several $\mathfrak{G}$-equivariant and $\mathfrak{G}$-invariant functions, they are themselves $\mathfrak{G}$-equivariant (in the case of $\mathbf{p}^n$ and $z^n$) and $\mathfrak{G}$-invariant (in the case of $r^n$ and $v^n$).

MuZero also computes returns starting from intermediate simulated state embeddings, $G(z^k)$, where $k$ is the depth of this state, assuming the actions simulated after this depth are $a^{k+1}, a^{k+2}, \ldots, a^{l+1}$. The return is also a $\mathfrak{G}$-invariant function, as it is a linear combination of several $\mathfrak{G}$-invariant functions:

$$
G(z^k) = \sum_{\tau=0}^{l-1-k} \gamma^\tau \rho_{\text{inv}}(z^{k+\tau}, a^{k+1+\tau}) + \gamma^{l-k} v_{\text{inv}}(z^l, a^{l+1}).
\tag{13}
$$

To prove that one planning step is equivariant, we need to show that the action selection is $\mathfrak{G}$-equivariant.

Since the outcome of MuZero's MCTS function is based on the initial observation, $s$, we denote the internal state of MCTS as $\{Q^s(z, a), N^s(z, a), \ldots\}$, for each node embedding $z$ that we simulate as we expand the tree, and each action $a$. These values correspond respectively to $Q$-values, visit counts, expansion policy estimates, etc. We will use identical notation as Schrittwieser et al. (2020) for these states, even though we denote the MuZero models, $h_{\text{eq}}, \tau_{\text{eq}}, p_{\text{eq}}, \rho_{\text{inv}}$ and $v_{\text{inv}}$, somewhat differently.

First, MuZero simulates a rollout by repeatedly selecting the next action to simulate, $a^k$, as follows:

$$
a^k = \arg\max_a \left[ Q^s(z^{k-1}, a) + P^s(z^{k-1}, a) \frac{\sqrt{\sum_b N^s(z^{k-1}, b)}}{1 + N^s(z^{k-1}, a)} \left( c_1 + \log\left( \frac{\sum_b N^s(z^{k-1}, b) + c_2 + 1}{c_2} \right) \right) \right],
\tag{14}
$$

where $c_1$ and $c_2$ are constant hyperparameters. Once the rollout is completed, the intermediate MCTS states are updated accordingly:

$$
\begin{aligned}
Q_t^s(z^{k-1}, a^k) &= \frac{N_{t-1}^s(z^{k-1}, a^k) Q_{t-1}^s(z^{k-1}, a^k) + G(z^{k-1})}{N_{t-1}^s(z^{k-1}, a^k) + 1} \\
N_t^s(z^{k-1}, a^k) &= N_{t-1}^s(z^{k-1}, a^k) + 1.
\end{aligned}
\tag{15}
$$

Note that we will, from now on, use shortened notation to represent group-transformed states, actions and embeddings. Specifically, we will use $\mathfrak{g}_s s$ as shorthand for $\rho_{\mathcal{S}}(\mathfrak{g})s$, $\mathfrak{g}_a a$ as shorthand for $\rho_{\mathcal{A}}(\mathfrak{g})a$, and $\mathfrak{g}_z z$ as shorthand for $\rho_{\mathcal{Z}}(\mathfrak{g})z$, in order to maintain brevity.

As discussed previously, we need to show that, for each MCTS internal state (e.g. $N^s$), if we assume $h_{\text{eq}}, \tau_{\text{eq}}, p_{\text{eq}}, \rho_{\text{inv}}$ and $v_{\text{inv}}$ to be $\mathfrak{G}$-equivariant functions, the resulting state would also be $\mathfrak{G}$-equivariant under transformations of the initial observation. That is, for all $s, a, z$:

$$
N^{\mathfrak{g}_s s}(\mathfrak{g}_z z, \mathfrak{g}_a a) = N^s(z, a).
\tag{16}
$$

To prove this, we will use induction on the number of backups performed by MCTS, $t$. We proceed:

$$
\begin{aligned}
\text{Base case } (t=0): N_0^{\mathfrak{g}_s s}(\mathfrak{g}_z z, \mathfrak{g}_a a) &= N_0^s(z, a) = 0 \\
Q_0^{\mathfrak{g}_s s}(\mathfrak{g}_z z, \mathfrak{g}_a a) &= Q_0^s(z, a) = 0.
\end{aligned}
\tag{17}
$$

Assume:

$$\text{Case } t : N_t^{\mathfrak{g}_s s}(\mathfrak{g}_z z, \mathfrak{g}_a a) = N_t^s(z, a)$$
$$Q_t^{\mathfrak{g}_s s}(\mathfrak{g}_z z, \mathfrak{g}_a a) = Q_t^s(z, a). \tag{18}$$

We will start by showing that the states and actions expanded by MCTS under initial $\mathfrak{G}$-transformed observation $\mathfrak{g}_s s$, $(\widetilde{z}^0, \widetilde{a}^1, \widetilde{z}^1, \widetilde{a}^2, \dots)$, would exactly correspond to $(\mathfrak{g}_z z^0, \mathfrak{g}_a a^1, \mathfrak{g}_z z^1, \mathfrak{g}_a a^2, \dots)$, where $(z^0, a^1, z^1, a^2, \dots)$ are states expanded under the non-transformed observation, $s$.

By equivariance of $h$, $\widetilde{z}^0 = h(\mathfrak{g}_s s) = \mathfrak{g}_z h(s) = \mathfrak{g}_z z^0$, as expected.

Next, we show that the actions selected by MCTS also obey a $\mathfrak{G}$-equivariance constraint, in the sense that: if $\widetilde{z}^{k-1} = \mathfrak{g}_z z^{k-1}$, then $\widetilde{a}^k = \mathfrak{g}_a a^k$.

As we assumed $N_t^s$ to be $\mathfrak{G}$-equivariant (Case $t$), it must hold that $\sum_b N_t^s(z, b)$ is $\mathfrak{G}$-invariant (as a sum-reduction of equivariant functions). Hence, we can rewrite Equation 14 as:

$$a^k = \arg\max_a \left[ Q_t^s(z^{k-1}, a) + P_t^s(z^{k-1}, a) \frac{\epsilon(z^{k-1})}{1 + N_t^s(z^{k-1}, a)} \right] \tag{19}$$

where $\epsilon$ is $\mathfrak{G}$-invariant, $P^s$ is $\mathfrak{G}$-equivariant by composition of functions that are $\mathfrak{G}$-equivariant by assumption, and $Q^s$ is $\mathfrak{G}$-equivariant by assumption of Case $t$.

Hence, using this formula to define $\widetilde{a}^k$, we recover:

$$
\begin{aligned}
\widetilde{a}^k &= \arg\max_a \left[ Q_t^{\mathfrak{g}_s s}(\widetilde{z}^{k-1}, a) + P_t^{\mathfrak{g}_s s}(\widetilde{z}^{k-1}, a) \frac{\epsilon(\widetilde{z}^{k-1})}{1 + N_t^{\mathfrak{g}_s s}(\widetilde{z}^{k-1}, a)} \right] \\
&= \arg\max_a \left[ Q_t^{\mathfrak{g}_s s}(\mathfrak{g}_z z^{k-1}, a) + P_t^{\mathfrak{g}_s s}(\mathfrak{g}_z z^{k-1}, a) \frac{\epsilon(\mathfrak{g}_z z^{k-1})}{1 + N_t^{\mathfrak{g}_s s}(\mathfrak{g}_z z^{k-1}, a)} \right] \\
&= \arg\max_a \left[ Q_t^{\mathfrak{g}_s s}(\mathfrak{g}_z z^{k-1}, \mathfrak{g}_a \mathfrak{g}_a^{-1} a) + P_t^{\mathfrak{g}_s s}(\mathfrak{g}_z z^{k-1}, \mathfrak{g}_a \mathfrak{g}_a^{-1} a) \frac{\epsilon(\mathfrak{g}_z z^{k-1})}{1 + N_t^{\mathfrak{g}_s s}(\mathfrak{g}_z z^{k-1}, \mathfrak{g}_a \mathfrak{g}_a^{-1} a)} \right] \\
&= \arg\max_a \left[ Q_t^s(z^{k-1}, \mathfrak{g}_a^{-1} a) + P_t^s(z^{k-1}, \mathfrak{g}_a^{-1} a) \frac{\epsilon(z^{k-1})}{1 + N_t^s(z^{k-1}, \mathfrak{g}_a^{-1} a)} \right] \\
&= \mathfrak{g}_a \arg\max_a \left[ Q_t^s(z^{k-1}, a) + P_t^s(z^{k-1}, a) \frac{\epsilon(z^{k-1})}{1 + N_t^s(z^{k-1}, a)} \right] \\
&= \mathfrak{g}_a a^k.
\end{aligned}
$$

We note that we have taken the $\mathfrak{g}_a$ out of the arg max, which is an unambiguous operation only if there is a unique action $a^k$ that maximises the expression in Equation 19. To avoid breaking the symmetry in practice, we propose that tiebreaks for $a^k$ are resolved in a purely randomised fashion.

Showing this, we now only need to verify that the updates to $N_t$ and $Q_t$ (in Equation 15) are equivariant for all state-action pairs along the trajectory. Values of $N$ and $Q$ for all other state-action pairs will be unchanged from $N_t$, and therefore trivially still $\mathfrak{G}$-equivariant.

First we show this for $N$:

$$
\begin{aligned}
N_{t+1}^{\mathfrak{g}_s s}(\widetilde{z}^{k-1}, \widetilde{a}^k) &= N_{t+1}^{\mathfrak{g}_s s}(\mathfrak{g}_z z^{k-1}, \mathfrak{g}_a a^k) \\
&= N_t^{\mathfrak{g}_s s}(\mathfrak{g}_z z^{k-1}, \mathfrak{g}_a a^k) + 1 \\
&= N_t^s(z^{k-1}, a^k) + 1 \\
&= N_{t+1}^s(z^{k-1}, a^k).
\end{aligned}
$$

Hence, Case $t+1$ still holds for $N$. Now we turn our attention to $Q$.

First, by invariance of $\rho$ and $v$, we can show that $G(z^k)$ is a sum of $\mathfrak{G}$-invariant functions and therefore also invariant. Plugging into the $Q$ update:

$$\begin{aligned}
Q_{t+1}^{\mathfrak{g}_s s}(\widetilde{z}^{k-1}, \widetilde{a}^k) &= Q_{t+1}^{\mathfrak{g}_s s}(\mathfrak{g}_z z^{k-1}, \mathfrak{g}_a a^k) \\
&= \frac{N_t^{\mathfrak{g}_s s}(\mathfrak{g}_z z^{k-1}, \mathfrak{g}_a a^k) Q_t^{\mathfrak{g}_s s}(\mathfrak{g}_z z^{k-1}, \mathfrak{g}_a a^k) + G(\mathfrak{g}_z z^{k-1})}{N_t^{\mathfrak{g}_s s}(\mathfrak{g}_z z^{k-1}, \mathfrak{g}_a a^k) + 1} \\
&= \frac{N_t^s(z^{k-1}, a^k) Q_t^s(z^{k-1}, a^k) + G(z^{k-1})}{N_t^s(z^{k-1}, a^k) + 1} \\
&= Q_{t+1}^s(z^{k-1}, a^k).
\end{aligned}$$

Hence, Case $t+1$ also holds for $Q$. The other states stored by MCTS, such as $P$, are computed by directly evaluating expressions in equation 12. As discussed before, these expressions are $\mathfrak{G}$-equivariant by composition.

Having proved that all internal state of of MCTS consistently remains transformed by $\mathfrak{G}$ under transformed input observations, we can conclude that the final policy given by MCTS will be exactly $\mathfrak{G}$-equivariant. $\square$

## 4 Experiments and results

### 4.1 Environments

We consider two 2D grid-world environments, MiniPacman (Guez et al., 2019) and Chaser (Cobbe et al., 2020), that feature an agent navigating in a 2D maze. In both environments, the grid-world state is represented as a 2D array (i.e. an image) and an action is a direction to move (one of $\{\rightarrow, \downarrow, \leftarrow, \uparrow\})^2$. Both of these grid-worlds are symmetric with respect to $90°$ rotations, in the sense that moving down in some map is the same as moving left in the $90°$ clock-wise rotated version of the same map. Hence, we take the symmetry group to be $C_4 = \{\mathfrak{e}, \mathfrak{r}_{90°}, \mathfrak{r}_{180°}, \mathfrak{r}_{270°}\}$, the 4-element cyclic group, which in our case represents rotating the grid-world state by all four possible multiples of $90°$.

### 4.2 Model hyperparameters

The architecture of all agents we study here is based on MuZero (Schrittwieser et al., 2020), with similar hyperparameters as in Hamrick et al. (2020). The encoder and the transition model are implemented as standard ResNet modules (He et al., 2016) with a hidden dimension of 128. Each ResNet starts with a $3 \times 3$ convolutional layer, followed by layer normalisation (Ba et al., 2016) and five residual blocks, each of them consisting of the following sequence of operations: layer normalisation, ReLU, $3 \times 3$ convolution, layer normalisation, ReLU and another convolution, to which the input of the residual block is added. After the residual blocks are applied, a final ReLU activation is used. For training the models used by MuZero, we maintain a prioritised experience replay buffer (Schaul et al., 2015), with batch size 512, discount factor $\gamma = 0.97$ and a trajectory length of $n = 10$ for computing $n$-step returns. All models are trained using the Adam SGD optimiser (Kingma & Ba, 2014) with learning rate $10^{-3}$. To prototype the use of equivariance in various parts of the MuZero models, we augment these ResNets in the way which is described in Section 3.1. Implementationally, this augmentation reduces to applying the ResNet on several $C_4$-transformed views of the input, and appropriately aggregating or concatenating the results as specified.

### 4.3 Results

We compare our $C_4$-equivariant implementation of EqMuZero (as described in section 3.1) with a standard MuZero that uses non-equivariant components: ResNet-style networks for the encoder and transition models, and MLP-based policy, value and reward models, following Hamrick et al. (2020). As the encoder and the policy of EqMuZero are the only two components which require knowledge of how the symmetry group acts

---

[2]Note that, to conform with the API of ProcGen, Chaser also contains some actions which do not transform equivariantly, though they also have no direct effect on the game state if played. We note that such actions will have equivariant effects in equivariant states, and hence our framework as presented can support their inclusion.

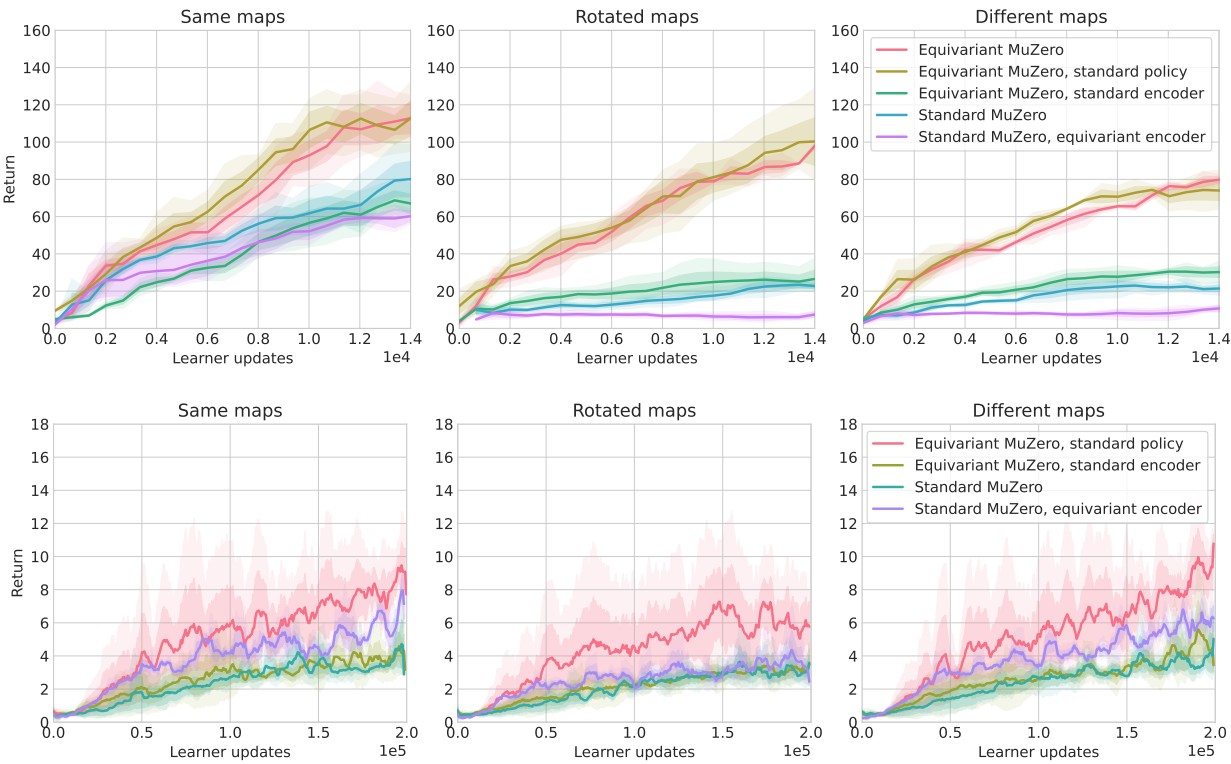

Figure 3: Results on procedurally-generated MiniPacman (top) and Chaser from ProcGen (bottom). The results are aggregated over three random seeds per each model variant. The dark shaded area is one standard deviation from the mean, and the light shaded area is the minimum/maximum value across the seeds.

on the environment, we include the following ablations in order to evaluate the trade-off between end-to-end equivariance and general applicability: Standard MuZero with an equivariant encoder, equivariant MuZero with a standard encoder and equivariant MuZero with a standard policy model.

We train each agent on a set of maps, $\mathbf{X}$. To test for generalisation, we measure the agent's performance on three, progressively harder, settings. Namely, we evaluate the agent on $\mathbf{X}$, with randomised initial agent position (denoted by *same* in our results), on the set of rotated maps $\mathbf{RX}$, where $\mathbf{R} \in \{\mathbf{R}_{90°}, \mathbf{R}_{180°}, \mathbf{R}_{270°}\}$ (denoted by *rotated*) and, lastly, on a set of maps $\mathbf{Y}$, such that $\mathbf{Y} \cap \mathbf{X} = \varnothing$ and $\mathbf{Y} \cap \mathbf{RX} = \varnothing$ (denoted by *different*).

Figure 3 (top) presents the results of the agents on MiniPacman. First, we empirically confirm that the average reward on layouts $\mathbf{X}$, seen during training, matches the average reward gathered on the rotations of the same mazes, $\mathbf{RX}$, for EqMuZero. Second, we notice that changing the equivariant policy with a non-equivariant one does not significantly impact performance. However, the same swap in the encoder brings the performance of the agent down to that of Standard MuZero—this suggests that the structure in the latent space of the transition model, when not combined with some explicit method of imposing equivariance in the encoder, does not provide noticeable benefits. Third, we notice that Equivariant MuZero is generally robust to layout variations, as the learnt high-reward behaviours also transfer to $\mathbf{Y}$. At the same time, Standard MuZero significantly drops in performance for both $\mathbf{Y}$ and $\mathbf{RX}$. We note that experiments on MiniPacman were done in a low-data scenario, using 5 maps of size $14 \times 14$ for training; we observed that the differences between agents diminished when all agents were trained with at least 20 times more maps.

Figure 3 (bottom) compares the performance of the agents on the ProcGen game, Chaser, which has similar dynamics to MiniPacman, but larger mazes of size $64 \times 64$ and a more complex action space. Due to the complexity of the action space, we only use EqMuZero with a standard policy, rather than a fully

equivariant version. We use 500 maze instances for training. Our results demonstrate that, even when the problem complexity is increased in such a way, Equivariant MuZero still consistently outperforms the other agents, leading to more robust plans being discovered.

Our theoretical analysis proves that the internal decision-making of MuZero's MCTS algorithm—and, hence, its derived visit counts—will be $\mathfrak{G}$-equivariant, provided its constituent networks are $\mathfrak{G}$-equivariant. However, as discussed in MuZero (Schrittwieser et al., 2020), the actions actually executed by the MuZero agent depend on stochastically sampling the *MCTS policy*, whose logits are based on these visit counts. As the action sampling is stochastic, executions of MuZero with different random seeds will yield different exact action sequences, leading to different downstream performance, even when the input maps are exactly rotated. Regardless, the simplification of the function search space for MuZero's neural models arising by equivariance constraints can indeed bring about clear improvements in performance, both for rotated maps and entirely unseen ones. This is especially the case in simpler, lower-data settings such as the MiniPacman environment. In the Chaser environment, the agent is exposed to a significantly visually richer input and a much larger dataset of maps. We posit that, in such a setting, rotated maps do not pose a substantially easier challenge than any other unseen map, yet still the equivariant versions of MuZero are capable of outperforming all other baselines we tested.

## 5 Limitations and future work

While the theory of Equivariant MuZero generalizes to any symmetry group, in this work we test an instance where the component neural networks satisfy the criteria for the $C_4$ group. Leveraging neural networks equivariant to continuous rotations, such as the SO(3) group, makes EqMuZero applicable to different problems, such as molecular tasks. However, for parts such as the encoder, the transition model and the policy, a different strategy would be required, as it is impossible to transform the input state and action with every element of an infinite group. Future work could consider enforcing the equivariance constraints via additional losses, possibly combining with an approach such as that of Park et al. (2022), keeping in mind that the theoretical guarantees will no longer apply in their current form. More generally, this work can also be composed with a module that discovers symmetries, such as in the work of Yang et al. (2023).

It is important to note the key benefit of enforcing equivariance within a world model: it reduces the hypothesis space of our world model to only functions respecting the relevant symmetries. This can, clearly, only improve the data efficiency of learning the world model, and when the complexities of the RL problem align nicely with these symmetric behaviours, then our approach can be highly beneficial for downstream performance. That being said, we acknowledge that there exist tasks requiring *hard exploration* (such as Montezuma's revenge), where the agent is tasked with discovering long sequences of actions, before any reward can be observed. Furthermore, these action sequences may not be amenable to the same symmetries as the underlying environment (horizontal reflections in the case of Montezuma's revenge). EqMuZero's innovations are not sufficient to resolve such challenges, but they are likely to be complementary with any exploration-based research that is conducted in the future.

## 6 Conclusions

We present Equivariant MuZero, a model-based agent that is, by construction, equivariant. We theoretically verify its properties with respect to general symmetry groups, proving the agent's overall equivariance given the appropriate conditions are met by its constituent neural networks. Moreover, we empirically demostrate that an Equivariant MuZero agent that is $C_4$-equivariant generalizes to unseen rotations of the training data, and also performs more robustly on test mazes, with diminishing returns when presented with 100 times more data.

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
