# OpenReview forum: "Equivariant MuZero"
_TMLR — Accepted by TMLR_

### Review · Reviewer_XLFJ · 2023-09-24

**Summary Of Contributions:**

The submission proposes a version of the MuZero algorithm equivariant to the C4 group, e.g. the group of the 4 90-degree rotations. This is useful in gridworlds which are often equivariant in this way (except e.g. when gravity is present such as in atari games). Experiments show the method was implemented correctly.

**Audience:**

Yes

**Broader Impact Concerns:**

\-

**Claims And Evidence:**

Yes

**Requested Changes:**

The submission claims that
>  ... agents need to learn the underlying rules governing the environment, so as to robustly generalise to conditions that differ from those they were trained on.

However, the proposed agent fails to do this. Instead of learning the underlying rules of the environment, the rules are hard-coded. Please clarify how this sentence relates to the proposed agent.

**Strengths And Weaknesses:**

Strengths
- The submission is technically correct.

Weaknesses
- The submission is limited in applicability to toy environments and is way behind state of the art, which uses more general theoretical treatment (van der Pol'20, Bronstein'21)

---

> ### Author Response · Authors · 2023-10-23
> **Reply to reviewer XLFJ**
>
> We thank the reviewer for taking the time to review our work and pointing out several suggestions for improvement.
>
> We would like to start by supplementing the reviewer’s summary of our contributions:
>
> > The submission proposes a version of the MuZero algorithm equivariant to the C4 group
>
> We would like to stress that the key contribution of our work is studying the effects of making MuZero’s constituent neural networks equivariant or invariant to a given symmetry group. We theoretically demonstrate that this causes the tree-search subroutine of MuZero to also act in an equivariant manner, and empirically show improved sample efficiency in two specific implementations over the C4 group. To the best of our knowledge, our work is the first one to both prove and demonstrate the utility of combining equivariance with Monte-Carlo tree search, which is one of the most influential model-based reinforcement learning methods.
>
> With this in mind, we answer one the reviewer’s remarks:
>
> > The submission is limited in applicability to toy environments and is way behind state of the art, which uses more general theoretical treatment (van der Pol'20, Bronstein'21)
>
> The C4 equivariant MuZero is, as explained above, only an illustrative implementation. Our theoretical framework is not limited to this setting, as it works for any G-equivariant architecture. Hence, Equivariant MuZero as presented can directly leverage any advances in equivariant NNs, such as those described in Bronstein’21 and van der Pol’20, both cited in our work.
>
> We have now also expanded the discussion at the end of Section 3.1 to make this point explicit and references state-of-the-art approaches for G-equivariant architectures that could be used in the framework of Equivariant MuZero.
>
> Regarding the reviewer’s comment:
>
> > Instead of learning the underlying rules of the environment, the rules are hard-coded.
>
> We’d like to point out that “learning rules” does not just mean learning the symmetries of the world, but learning the entire world-model dynamics. Therefore, hard-coding the symmetries of a world model is not sufficient to fully specify the rules of the environment and the agent still has to learn those rules. Our work demonstrates that hard-coding the symmetries into a world-model makes those rules easier to learn for an agent, as otherwise it would have to learn the symmetries directly from data, and world-models trained with hard-coded symmetries support significantly better decision-making with MCTS.
>
> We have now amended the abstract to clearly decouple what is learnt (a world-model) and what is hard-coded (symmetries).
>
> We hope that our responses address your comments accordingly and look forward to further discussions with you.

---

### Review · Reviewer_QDcZ · 2023-10-08

**Summary Of Contributions:**

This paper studies the problem of improving sample complexity and generalization of model-free method -- specifically MuZero -- by taking advantage of symmetries in the environment. To do so, it introduces architectural modifications in the networks that MuZero is composed of, such that the internal computations of state embeddings, transitions, and the policy, are equivariant, and the reward and Q-value computation is invariant, to the symmetries in the environment.
The paper proves that these architectural modifications lead to the actions taken by MuZero itself being equivariant to symmetries in the environment.
The experiments to validate this equivariant architecture are conducted in two domains, MiniPacman, and Chaser from the ProcGen suite of domains. These experiments seek to show that MuZero learns faster with the equivariant architecture, and the subsequent policies generalize better to symmetrically transformed maps in these games as well as novel maps that the agents have never trained on before.

**Audience:**

Yes

**Broader Impact Concerns:**

Principled ways to improve sample complexity and generalization of RL methods such as these would be helpful to ensure consistent and expected behavior of RL agents. I would imagine the broader impacts of these investigations to be mostly positive.

**Claims And Evidence:**

Yes

**Requested Changes:**

Some minor changes that would be good to have:
  * The paper specifies that equivariant actions are also needed for the symmetries to hold very late, in Section 3.1. Perhaps introduce or clarify it earlier?
  * The paper mentions that the Chaser environment includes some non-equivariant actions. Could you mention how those are handled explicitly? I assume you do not need to do anything special since the same actions should have equivariant effects in equivariant states. But explicitly stating this effect will be helpful.

**Strengths And Weaknesses:**

## Strengths:
* The paper has a clear hypothesis: incorporate symmetries in the environment into the MuZero architecture, and show that this modification leads to faster learning and better generalization.
* The modifications made to MuZero to take advantage of a particular kind of symmetry ($C_4$-equivariance), are well thought out.
* The above modifications ground the theoretical analysis that shows that MuZero will take equivariant actions in equivariant states if corresponding modifications are made for any equivariance, which in itself seems to be correct (see comments below)
* The experiments validate the above hypothesis that MuZero learns faster and generalizes to new instances of the environment better when the equivariant components are incorporated.
* The ablations also make sense and give confidence that all the architectural changes made are valuable, and which ones are essential.

## Weaknesses
* While the experiments show that EqMuZero can take advantage of symmetries in the environment, a question that remains is how well it would deal with scenarios where the symmetries are not a major component of what makes a problem difficult (such as left-right symmetry in say Montezuma's revenge, with a lot of actions that are not part of the equivariance, and where the exploration problem is hard regardless of the symmetries)
* The other points of contention are fairly minor, and I mention them here with questions I have:
  * In the background section, functions and sets are notated with the same format (capital letters).
  * An action encoder is defined to map actions from $A \longmapsto Z_A$, but the transition and reward functions do not take advantage of this encoding.
  * In Figure 1, shouldn't the space after decoding be the same space as before  encoding? $S$ instead of $\mathbb{R}^l$ in this case?
  * In proof of Theorem 1, on page 8 when deriving $\tilde{a}^k = g_a a^k$, I am unsure of how the last two steps are arrived at. Specifically, how $g_a$ is taken outside the $argmax$.

---

> ### Author Response · Authors · 2023-10-23
> **Reply to reviewer QDcZ**
>
> We thank the reviewer for a very constructive review that recognises our work’s key contributions and offers many avenues for further discussion.
>
> Regarding the comment:
>
> > a question that remains is how well it would deal with scenarios where the symmetries are not a major component of what makes a problem difficult
>
> We fully agree with this point and reiterate that while Equivariant MuZero will provably help data-efficiency under symmetries, this may not be sufficient for tackling hard-exploration problems, especially in situations such as the ones you highlight. We have now added a paragraph in the future work section which emphasises this as a fruitful direction of future study.
>
>
> We will address the next points in order:
>
> > In the background section, functions and sets are notated with the same format (capital letters).
>
> We understand this as referring to the MDP notation (S, A, P, R, γ), where P and R are functions denoted by capital letters. Could you confirm that this is the contentious part of the notation? We are happy to modify it if so.
>
> > An action encoder is defined to map actions from $A \rightarrow Z_A$ but the transition and reward functions do not take advantage of this encoding.
>
> Indeed, this hasn’t been explicitly addressed in the manuscript – the transition and reward functions internally call the action encoder to embed the actions passed to them. Equation 7 illustrates this, but as this is quite late in the paper, we have now added a footnote immediately when the transition and reward models are first introduced, drawing attention to this fact.
>
> > In Figure 1, shouldn't the space after decoding be the same space as before encoding?
>
> We use the term decoder as a general component which predicts certain properties of the state. While auto-encoders are indeed one possible realisation of this, we explicitly highlight other options in the last paragraph of 2.5, such as “taking the decoder to be the reward model and l = 1” and “the policy model as the decoder and l = |A|”.
>
> > I am unsure of how the last two steps are arrived at
>
> Our argument is based on the observation that actions appear in the argmax only transformed by $\mathfrak{g}_a^{-1}$. That is, we are looking at a maximisation of the form $$\arg \max_a f(\mathfrak{g}_a^{-1}a)$$ and therefore the action optimising this expression is exactly the one that, when $\mathfrak{g}_a^{-1}$-transformed, optimises $f(a)$.
>
> And since $\mathfrak{g}_a\mathfrak{g}_a^{-1}a = a$ for all $a$ by definition, $$\arg \max_a f(\mathfrak{g}_a^{-1}a) = \mathfrak{g}_a\arg \max_a f(a)$$ justifying our first step.
>
> Note that, as we mentioned in the manuscript, this step rests on the assumption that there is exactly one action ($a^k$) optimising $f(a)$; we proposed resolving tiebreaks in a randomised fashion to avoid symmetry breaking in practice.
>
>
> > The paper specifies that equivariant actions are also needed for the symmetries to hold very late, in Section 3.1. Perhaps introduce or clarify it earlier?
>
> Thank you for your suggestion, we have now explicitly mentioned this in section 2.4.
>
> > I assume you do not need to do anything special since the same actions should have equivariant effects in equivariant states
>
> This is correct and we have now made it explicit.
>
> We hope that our responses address your comments accordingly and look forward to further discussions with you.

---

> > ### Comment · Reviewer_QDcZ · 2023-11-08
> > **Thank you for the response**
> >
> > The author responses have addressed my questions.
> > I thank the authors for clarifying my doubts.
> >
> > > We understand this as referring to the MDP notation (S, A, P, R, γ), where P and R are functions denoted by capital letters. Could you confirm that this is the contentious part of the notation? We are happy to modify it if so.
> >
> > Yes. This section is what I was referring to. Modifying to separate the functions and sets by use of some other font for one of them would be helpful.

---

### Review · Reviewer_GAxJ · 2023-10-09

**Summary Of Contributions:**

The paper introduces a new version of the MuZero algorithm, called Equivariant MuZero (EqMuZero), which explicitly incorporates the symmetries of the environment into its world-model architecture. The main claims of the paper are:

1. The authors propose a new approach to improve the data efficiency and generalization capabilities of MuZero by incorporating the symmetries of the environment into its world-model architecture. This approach is based on the concept of equivariance, which ensures that the action-selection algorithm of the EqMuZero agent will behave consistently across equivalent states.
2. They provide a proof that guarantees that if all neural networks used by MuZero are equivariant to a particular symmetry group acting on the environment, all actions taken by the agent will also be equivariant to that same symmetry group. This means that EqMuZero can be more data-efficient than standard MuZero, as it knows how to act in states it has never seen before.
3. The paper also presents a specific implementation of EqMuZero that is equivariant to 90-degree rotations by design. The authors claim they will test this implementation on two rotationally symmetric grid-worlds: procedurally-generated MiniPacman and the Chaser game in the ProcGen suite, to verify the generalization capabilities of EqMuZero, demonstrating the benefits of incorporating equivariance into the world-model architecture.
4. The authors also claim that their performance improvements hold even when only some of the components of Equivariant MuZero obey strict equivariance, which would show the robustness of their construction.

**Audience:**

Yes

**Claims And Evidence:**

No

**Requested Changes:**

- Add related works and discuss them: use aforementioned papers to provide a more holistic understanding of the current landscape of equivariant neural networks, symmetry in planning/search, and model-based RL.
- Attempt to employ existing equivariant neural network techniques (e.g., E(2)-steerable CNN by Weiler et al.) as done in Zhao et al. (ICLR 2023), or convincing explanation/comparison.
- A thorough examination on how $D_4$ symmetry could be handled within the author’s proposed framework is useful. It could provide improve sample efficiency, which is the core claim of using equivariance in the paper.
- A more comprehensive analysis of the equivariance in MuZero is urged, especially on using equivariant components in different places, given it's the key of the paper’s contribution.

**Strengths And Weaknesses:**

Strengths:

- The paper explores an important idea: improving data efficiency in MuZero using equivariance, which is a new and relevant approach.
- A clear plan is proposed to achieve equivariance in MuZero, which hasn't been done before for MuZero-based approaches, making this a notable contribution.
- The paper provides a strong guarantee that the new EqMuZero will act consistently in similar situations, backing up its claims to improve data efficiency and generalization.
- The specific way EqMuZero is designed to handle 90-degree rotations is smart, especially for the chosen testing environments. This shows a good attempt to prove their claims.
- The paper mentions that their approach stays strong even when not all parts of Equivariant MuZero follow strict equivariance, hinting at potential usefulness in real-world scenarios with imperfect symmetry.

Weaknesses and Concerns:

- The paper falls short in acknowledging an entire spectrum of work on equivariant neural networks. Using the latest techniques, the paper will be much more interesting, instead of limited to $C_4$.
    - They have been used quite intensively in most of the equivariant model-free/model-based RL work. For this paper to bring enough interests to the field, it would be convincing to do any of the following (1) give convincing reasons why missing the entire body of works on equivariant NNs, (2) give empirical comparison that the proposed approach is better in some sense, (3) show this approach is technically interesting via other evidences.
    - The approach in Sec 3.1 seems a special case of G-CNN and Steerable CNN — it can only consider regular representation of $C_4$ (cyclic permutation of feature channels), while it didn’t consider other representations (standard, other irreducible representations) or groups (dihedral groups). This specific implementation resembles the group equivariant convolution network (G-CNN, Cohen et al. 2016), which should be acknowledged. It is the backbone of most current equivariant network architecture. Cohen et al. later developed Steerable CNN and extended G-CNN, which can implement all these features.
    - The proposed method seems to not be able to handle this case: Equivariant mapping from regular representation (4 channels indicating 4 directions) to standard representation (output is a 2D vector, e.g., using (0,1) to represent north). Such equivariance constraint needs to solve the space of intertwiner / steeable constraint of $W_{2 \times 4}\rho_{reg}(g) = \rho_{std}(g) W_{2 \times 4}$, where the existing work has already provided solutions, see E(2)-steerable CNN by Weiler et al. (ICLR 2021).
- There is absence of closely related work concerning symmetry in planning/search and model-based reinforcement learning (RL).
    - For instance, the paper "Integrating Symmetry into Differentiable Planning with Steerable Convolutions" by Zhao et al. (ICLR 2023) delves into $D_4$ symmetry, which could have offered a richer contextual understanding.
    - The paper "EDGI: Equivariant Diffusion for Planning with Embodied Agents" by Brehmer et al. (arxiv 2023), could have provided a broader perspective on equivariant diffusion models for equivariant model–based planning.
    - Furthermore, the paper "Can Euclidean Symmetry Help in Reinforcement Learning and Planning?" (arxiv 2023) discuss how a model-based RL algorithm can be made equivariant and could serve as a pertinent reference, although it's uncertain to what extent it’s related.
- The paper is not using the full symmetry of a planar board $D_4$ dihedral group (4 rotations times 2 reflections = 8 transformations), as shown in Zhao et al. 2023. Can the author’s approach handle D_4?
- There are some other technical concerns. For example, the policy probability $\pi(a|s) = \pi(g.a|g.s)$  should be invariant, while $\pi(s)=a$ is equivariant. Also, why not using fully equivariant policy? The theoretical part only showed for equivariant version.

---

> ### Author Response · Authors · 2023-10-23
> **Reply to reviewer GAxJ (Part I)**
>
> We thank the reviewer for a detailed review which recognises our key contributions and offers many helpful pointers to current geometric deep learning advances.
>
> We first address your comments regarding relevant equivariant neural network developments.
>
> We would like to reiterate that the key contribution of our work is demonstrating, theoretically as well as empirically, that equivariant neural networks compose appropriately with Monte-Carlo tree search as employed by MuZero.
>
> As such, the aim of our paper’s empirical evaluation is to demonstrate, on a concrete example (C4 in our case) that equivariant neural networks improve the data efficiency of MuZero—not to perform a comprehensive comparison of possible equivariant network designs.
>
> Our specific C4-equivariant MuZero is thus only an illustrative implementation. As your review recognises, our implemented architecture is a special case of G-CNNs and its current formulation will not generalise to more complicated groups. In the paper as submitted, we explicitly pointed out this limitation in section 5, and we have now cited Cohen et al. in several places around the document to make the link to G-CNNs explicit.
>
> Our theoretical framework is, of course, not limited to this setting, and it works for any G-equivariant architecture. We have now expanded the related work discussion in Section 2.4 to discuss state-of-the-art approaches for G-equivariant architectures (including Steerable CNNs) that could be used in the framework of Equivariant MuZero.
>
> We now turn attention to your remarks about prior connections of symmetry and MBRL. We find all of the papers you suggested to be relevant to ours, and we now discuss all of them appropriately in the related work (Section 2.5.).
>
> > "Integrating Symmetry into Differentiable Planning with Steerable Convolutions" by Zhao et al. (ICLR 2023)
>
> This work incorporates G-equivariance into Value Iteration Networks (VINs), which are directly inspired by algorithmic alignment to a well-known MBRL algorithm, Value Iteration. However, VINs are _implicit planners_; while they are inspired by a planning method, they are realised as end-to-end differentiable, model-free agents. Our work, instead, theoretically and empirically studies the utility of combining equivariance with an actual search algorithm (MCTS).
>
> > "EDGI: Equivariant Diffusion for Planning with Embodied Agents" by Brehmer et al. (arxiv 2023)
>
> This work incorporates G-equivariance into diffusion-based planning (Janner, 2022). Here the model is asked to perform planning in a trajectory space induced by a diffusion model; that is, by iteratively refining randomly-sampled noise. It is indeed a very interesting perspective on planning, which also stands to benefit from equivariant modelling, and we now discuss it.
>
> > "Can Euclidean Symmetry Help in Reinforcement Learning and Planning?" (arxiv 2023)
>
> We first note that this work is definitely concurrent to ours, as it has appeared on the arXiv fairly recently. It has traits that are indeed relevant to our work, studying how various path-planning based algorithms (e.g. MPC-based) behave under symmetry transformations, and derive arguments that are similar in spirit to our theoretical analysis of MuZero under equivariance. As such, it is indeed worthwhile to reference this work, and we have done so in the related work.

---

> ### Author Response · Authors · 2023-10-23
> **Reply to reviewer GAxJ (Part II)**
>
> On your remark concerning D4 symmetry:
>
> > A thorough examination on how D4 symmetry could be handled within the author’s proposed framework is useful.
>
> We have now added a remark on this in Section 3.1. Since D4 incorporates reflections as well as rotations, we would need to extend our representation to handle eight views rather than four. Of course, having this many views might prohibit the latent state dimension due to memory constraints, and in such cases we would recommend using approaches such as Steerable CNNs—as employed by Zhao et al.
>
> Regarding analyses of equivariant components in EqMuZero, we argue that the ablative tests in Figure 3 already study the effect of removing equivariance in various places (e.g. making the encoder non-equivariant, making the policy head non-equivariant, or making all but the encoder non-equivariant). These experiments demonstrate that, to reap the full benefits of data efficiency on the environments we study here, it is important to have most of the neural components (encoder, transition, value and reward) be equivariant or invariant, whereas removing equivariance in the policy network did not yield significant differences.
>
> Concerning the equivariance of the policy network:
>
> > For example, the policy probability $\pi(a|s)=\pi(g.a|g.s)$ should be invariant, while $\pi(s)=a$ is equivariant. Also, why not using fully equivariant policy? The theoretical part only showed for equivariant version.
>
> It is true that the policy probability of a _particular action_ is designed to be invariant to a g-action. That being said, when we say a policy is equivariant, we refer to the full output of a policy network—not just one logit, but a vector of logits of size $|\mathcal{A}|$. Hence, in the actual implementation (as exemplified by Figure 2), we indeed use a fully equivariant policy.
>
> For ProcGen, a more complex action space is present, which would have required a more elaborate equivariant policy implementation. Given this, and the fact that our MiniPacman experiments already indicated that making the policy non-equivariant does not yield weaker data efficiency, and the overall higher computational demands with running ProcGen experiments, we chose to focus only on the standard policy network for the ProcGen experiments.
>
> We hope that our responses address your comments accordingly and look forward to further discussions with you.

---

> ### Comment · Reviewer_GAxJ · 2023-11-08
>
> I appreciate the authors' efforts in addressing my concerns and questions. I find some concerns addressed, such as the discussion of equivariant NNs and equivariant model-based RL and some technical questions, while some other ones still need more work on the paper itself.
> For example, for using existing equivariant NN architectures, one benefit is that it supports many group representations (trivial, standard, regular, etc) and can be used on more diverse set of tasks, while the current set of tasks seems to be limited. Another benefit is that it can easily extend to D4 or other groups.
> Overall, this work seems valuable to me while the empirical component could still be improved to match the claims of the paper.

---

### Decision · Action_Editor_zMve · 2023-11-14

**Recommendation:** Accept with minor revision

**Comment:**

This paper studies how symmetries can be exploited to improve the efficiency of model-based RL methods.

All the reviewers agree that the contribution is interesting, albeit narrow in its scope, as it only focuses on the MuZero algorithm and provides empirical evidence for one specific group symmetry.

Whereas the experimental evalaution looks to be sound, it is somewhat limited and could be improved (see requested changes below).

I report below some changes that are necessary to clear the acceptance of the paper and some additional suggestions to further improve the experimental analysis.

Requested changes (**necessary for acceptance**):
- It is not imemdiately clear from the text how many indipendent runs have been considered to generate the plots nor the meaning of the shaded area around the curves. Please specify those in the text and/or the figure captions;
- Include a deeper discussion of the reported results, e.g., why the performance for *different* maps looks to be better than *same* in the Chaser domain? Instead, why the performance declines slightly in *rotated* maps despite the equivariance?
- Report detailed experiments setup and how to reproduce the results (possibly in the Appendix).

Suggestions (**optional**):
- Consider reporting the "number of samples" instead of the "learner updates" in the x-axis of Figure 3;
- Consider releasing the code used to run the experiments, which can benefit future research on the same topic (e.g., similar experiments with more general equivariant architectures).

**Audience:**

The paper is likely to draw the interest of the model-based RL community.

**Claims And Evidence:**

The claims of the paper are supported both with theoretical derivations and empirical evidence. However, the experimental campaign could be improved.

---

> ### Author Response · Authors · 2023-12-18
> **Thank you! Camera-ready revision uploaded.**
>
> We would like to thank the Action Editor for handling our paper, conveying the positive decision and for the additional comments!
>
> This message is to confirm that we have just submitted the final camera-ready version of the paper, taking into account your requested changes. Specifically: we provide the necessary clarifications in the caption of Figure 3, more deeply discuss the relative order of the studied models in the end of Section 4.3., and provide more details on the hyper-parameters and experimental setup in Section 4.2.
>
> Best,
> Authors

---

> > ### Comment · Action_Editors · 2023-12-19
> > **Ready for publication**
> >
> > Dear Authors,
> >
> > Thank you for taking care of the requested changes. The paper is now ready to be published by TMLR. Congratulations!
> >
> > Finally, I want to highlight below a comment from the private discussion with Reviewer GAxJ, who signalled directions to improve the work. While I do not agree with the Reviewer that those changes must be implemented in this paper to clear the bar for acceptance, I still believe this is a very valuable feedback for the authors or whoever wants to pursue advancements on this research line.
> >
> > Best,
> > Action Editor
> >
> >
> > -----
> >
> > Reviewer GAxJ wrote:
> >
> > 1. **Engagement with Current Research on Equivariant Neural Networks:** The paper should thoroughly acknowledge and integrate recent advancements in equivariant neural networks to ensure its relevance and depth. It should at least discuss why they did not even compare to them.
> > 2. **Expansion of Experimental Evaluation:**
> >    - **Incorporation of Complex Applications:** The current experimental setup is limited to relatively simple environments. A more compelling evaluation could include complex, real-world applications such as the game of Go, which would provide a rigorous test of the proposed methods in a challenging and relevant context.
> >    - **Comparative Analysis with Advanced Techniques:** Implementing and comparing results with recent techniques like E(2)-steerable CNNs could offer a robust benchmark for the proposed approach.
> > 3. **Better Review of Related Literature:** (Optional) I'm happy to see more related works beyond a few I mentioned, such as more in MDP homomorphism for symmetry.
> > 4. **Detailed Examination of D_4 Symmetry and Technical Improvements:** An in-depth analysis of how the paper's methodology handles D_4 symmetry is crucial. C_4 is not full symmetry of the environment and is pretty easy to implement, which has been well addressed before 2016. Additionally, addressing the technical concerns raised, especially regarding policy probability and the application of fully equivariant policies, is necessary for a holistic understanding.
> >
> > I identify two potential areas of contribution: (1) Equivariance: The current treatment of equivariance appears quite constrained and does not fully utilize the techniques outlined in Cohen et al. 2016. Enhancing this aspect could substantially enrich the paper's technical depth. (2) Equivariance in Model-Based RL: While this topic has been explored in prior works, a more nuanced and detailed exploration within this paper could be beneficial. The most notable contribution appears to be the application of equivariance in MuZero. For this, I would like to see a more tailored and in-depth description, accompanied by a comprehensive empirical study that demonstrates its effectiveness and distinct advantages. This focus could significantly strengthen the paper's overall impact and originality.